# The Inlay Technique with Cortico-Cancellous Olecranon Bone Graft Used for Revision of Failed Distal Interphalangeal Joint Arthrodesis

**DOI:** 10.3390/medicina58101442

**Published:** 2022-10-13

**Authors:** Sang-Hyun Ko, Joo-Won Park, Tong-Joo Lee

**Affiliations:** Department of Orthopaedic Surgery, Inha University Hospital, Incheon 22332, Korea

**Keywords:** hand, finger joint, arthrodesis, bone graft

## Abstract

*Background and Objectives*: Although distal interphalangeal (DIP) arthrodesis is an effective surgical method for end-stage osteoarthritis of the phalangeal joint, the nonunion rate of DIP arthrodesis has been reported to range from 15% to 20%. To this end, we devised an inlay technique with a cortico-cancellous olecranon bone graft for failed DIP arthrodesis. This study aimed to introduce the inlay bone grafting technique for failed arthrodesis of the DIP joint and demonstrate its advantages. *Materials and Methods*: We reviewed consecutive 19 digits (15 patients) who had undergone DIP revision arthrodesis using the technique at our institution between January 2010 and December 2020. The observed outcome measures were the bone union rate, and related complications. Bone union was evaluated using follow-up radiography. The quick Disabilities of the Arm, Shoulder and Hand (DASH), visual analog scale (VAS) for pain, and VAS for satisfaction assessed patient function and perceived clinical outcomes. *Results*: No major complications were observed at the recipient site. The average VAS for pain and satisfaction and DASH score improved from preoperatively to 6 months after surgery (both, *p* = 0.001). *Conclusions*: The inlay technique with cortico-cancellous olecranon bone grafts showed excellent bone union rates and functional scores with nonunion of the DIP joint. This technique may be an adequate surgical option for patients with confirmed nonunion of the DIP joint and persistent symptoms.

## 1. Introduction

Arthrodesis of the distal interphalangeal (DIP) joint (thumb interphalangeal joint) is the most reliable surgical treatment for painful arthritis when all non-operative methods have failed [1]. Since Bunnell first introduced a technique for DIP arthrodesis in 1955, consisting of resection of the articular surfaces and securing of the joint with Kirschner (K) wires, a variety of arthrodesis stabilization techniques, including interosseous wiring, tension band wiring, plate fixation, or variable-pitch cannulated headless screws such as Herbert (Zimmer, Inc., Warsaw, IN, USA) or Acutrak (Acumed Inc., Beaverton, OR, USA) have been reported [2,3,4,5].

DIP arthrodesis is an effective surgical method; however, it has a high complication rate, which remains a challenge. The nonunion rate of DIP arthrodesis has been reported to range from 15% to 20% [6,7]. Common reasons for nonunion include poor bone stock, resection of subchondral bone, soft tissue damage, premature pin removal, and infection [6,7,8]. There is limited research on the nonunion of DIP joint arthrodesis and various factors, including DIP joint complexity, joint stiffness, and soft tissue conditions, which makes it difficult to select a surgical method for revision. Considering these problems, we have operated on patients with failed DIP arthrodesis using the cortico-cancellous bone graft inlay technique since 2010. However, to date, no study has assessed the effects of this technique.

This study aimed to introduce the inlay bone grafting technique for failed arthrodesis of the DIP joint and demonstrate its advantages. We hypothesized that this technique would simultaneously provide good finger function restoration and reasonable union rates.

## 2. Materials and Methods

### 2.1. Patient Selection

We retrospectively reviewed patients with failed DIP arthrodesis who were treated with the inlay bone grafting technique at our institution between January 2010 and December 2020. The study subjects included patients with failed DIP arthrodesis who were followed up for at least 6 months after revision surgery. The diagnosis of failed DIP arthrodesis was made based on clinical and radiologic assessments by an orthopedic surgeon at the outpatient visit. Indications for DIP joint arthrodesis revision include persistent pain or tenderness and no progression toward healing over 3 consecutive months on serial radiography [9]. Exclusion criteria included postoperative finger injury, no follow-up for more than 6 months postoperatively, and diabetes or immunosuppression [10].

The design and protocol of this study were reviewed and approved by the Institutional Review Board (IRB) of our institution (IRB No. INHAUH 2021-08-021). Verbal informed consent was obtained from all participants before the study. The IRB waived the requirement for written informed consent because of the study’s retrospective nature using medical records.

### 2.2. Surgical Procedure

An experienced orthopedic surgeon performed all surgeries. All patients underwent surgery under axillary regional anesthesia. A lazy S-shaped skin incision was made over the dorsal aspect of the DIP joint to expose the nonunion site. Scar, fibrous tissue, osteophytes, and calluses were removed using curettes, rongeurs, and surgical burrs. Internal fixators such as intramedullary screws were removed. During DIP joint flexion, the nonunion site was reached through cuts with a saw blade in all patients. Holes were made at the nonunion site using a surgical burr (2 mm in diameter, round), and the hole size was measured for bone grafting (Figure 1A). Then, a 1.5 cm incision was made distal to the tip of the olecranon for bone harvest. An incision was made through the periosteum, which was dissected off the donor site (Figure 1B). The cortico-cancellous bone graft was harvested using an osteotome and a small oscillating saw (Figure 1C). The bone peg was impacted into the hole created in the middle phalanx with DIP joint flexion (Figure 1D), and the joint was extended such that the distal aspect of the bone peg fitted into the opposite side of the distal phalanx hole. (Figure 1E) The bone graft was shaped at the nonunion site and fixed using K-wires (0.8–1.2 mm) (Figure 1F) [6,11]. The finger was protected with a splint, and a compression dressing with an elastic band was used on the elbow. Patients were allowed to move their elbow freely.

### 2.3. Follow-Up and Postoperative Rehabilitation

The DIP was protected using a splint for 4 weeks. Proximal interphalangeal and metacarpophalangeal joint range of motion exercises were initiated during this interval. Postoperatively, patient follow-up using radiographs at the outpatient clinic was performed at 2, 4, and 6 weeks, and every month thereafter. The splint was removed at the outpatient 4 weeks after surgery. If bone union was confirmed on radiography, the internal fixator, such as the K-wires, was removed on an outpatient basis under local anesthesia. Then, finger movements, such as pinch exercises, were allowed with the removal of internal fixators.

### 2.4. Clinical Assessment

Postoperative functional outcomes were assessed using a visual analog scale (VAS, 0–10 cm) for pain and satisfaction and the quick Disabilities of the Arm, Shoulder and Hand (DASH) at each follow-up appointment postoperatively. The preoperative and 6 months postoperative VAS and quick DASH scores were compared. Complications, such as infection, nonunion, and deformity after implant removal, were assessed in the outpatient clinic.

### 2.5. Radiological Assessment

Postoperative radiologic assessments were performed using the fingers’ anteroposterior, lateral, and oblique views. Nonunion of the DIP joint was defined as persistent pain or tenderness with no apparent potential to heal without intervention and did not show progression toward healing over 3 consecutive months on serial radiography [11,12]. Delayed union was defined in adults as a healing time of more than 12 weeks [13]. Radiography after revision surgery was serially checked to confirm the union and the time of union. All radiographs were evaluated separately by three orthopedic surgeons using an INFINITT M6 Picture Archiving and Communication System (INFINITT Healthcare Co., Ltd., Seoul, Korea).

### 2.6. Statistical Analysis

Categorical variables are presented as absolute numbers or means (± standard deviation). Clinical outcomes were compared using Wilcoxon signed-rank test in cases with non-normal distribution. Statistical significance was set at *p* < 0.05. Statistical analysis was conducted using IBM SPSS Statistics for Windows (version 25.0; IBM Corp., Armonk, NY, USA).

## 3. Results

A total of 107 digits (82 patients) with DIP joint arthritis who underwent DIP joint arthrodesis were retrospectively reviewed: 20 digits in 15 patients with failed DIP arthrodesis were treated using the inlay bone grafting technique. However, one digit in one patient was excluded from the study because they met the exclusion criteria and were lost to follow-up. Fourteen patients with failed DIP arthrodesis who met the study criteria were evaluated. Patient demographics are summarized in Table 1. There were 14 patients eligible for follow-up, 2 (14.3%) men and 12 (85.7%) women, with an average age of 56.6 years (minimum 42 years, maximum 69 years). The mean time between primary and secondary operation was 19.6 (±4.9) weeks (range, 16–32 weeks). The mean follow-up period was 29.6 (±4.3) weeks (range, 24–40 weeks). There were 15 cases using K-wires and 4 cases using headless screws at the primary surgery; however, all patients used the method with K-wires for failed DIP joint arthrodesis.

The mean preoperative VAS score was 4.29 (±1.33) for pain and 2.29 (±1.07) for satisfaction. After 6 months, each value significantly improved, to 2.14 (±0.77) and 6.21 (±1.12) (both, *p* = 0.001). The mean preoperative quick DASH score was 59.09 (±9.38), and 6 months after surgery, it was 14.92 (±6.03) (Table 2 and Figure 2). No major complications were observed at the recipient or donor sites. One patient had a superficial infection at the DIP arthrodesis site. The symptoms of infection improved with oral antibiotics for 1 week. No cases of skin necrosis or malunion were observed. There was no significant discomfort related to the surgical and donor sites in the other patient.

All patients had complete bone union at the final follow-up (Figure 3). The mean duration of bone union was 9.2 (±2.5) weeks (range, 7–16 weeks). One patient had delayed union, with 16 weeks of bone union.

## 4. Discussion

Failed arthrodesis of the DIP joint is a challenge. However, specific guidelines for the treatment do not exist, and no single technique has gained universal popularity [14]. Therefore, we contrived the inlay bone grafting technique in patients with failed arthrodesis of DIP to enhance the bone healing while improving stability. We hypothesized that the inlay technique with cortico-cancellous olecranon bone graft would simultaneously provide satisfactory finger function restoration and good union rates for failed arthrodesis. We had a 100% bone union rate at the final follow-up, and clinical outcomes with VAS and quick DASH scores improved from preoperatively to 6 months after surgery using the inlay technique. There were no major complications, including nonunion, infection, or malunion.

Nonunion of DIP joint was reported to occur in up to 20% of cases after fusion surgery, and it causes functional finger deterioration, affecting the postoperative outcome and rehabilitation [6,7]. Our study showed that 20 digits (18.7%) out of 107 digits had nonunion in primary DIP fusion, obtaining similar results to those of other studies. In comparison with other joints, revision surgery of the phalangeal joint, including the DIP joint, is complex, and the nonunion rate has been reported to be high because of the lack of bone stock and ligamentous stability of the joint [8]. Therefore, a more careful choice of surgical treatment should be made for failed nonunion to reduce the nonunion rate and complications.

Various surgical techniques for arthrodesis of DIP joint have been described in a series of articles, and K-wire fixation and compression screws are commonly used in joint arthrodesis operations [15]. Herbert screw fixation was found to have increased anteroposterior bending strength and torsional rigidity compared with tension band wiring and K-wire fixation [16]. The disadvantages of this technique are related to a learning curve, and several screws in that study were removed due to hardware prominence and misplacement of the screw [17]. It may be difficult to remove the compression screws in the outpatient setting, and bone defects or soft tissue damage may occur. Compared with compression screws, K-wires have been used for many years, leading to insufficient osseous stabilization and compression strength [1]. However, intramedullary K-wire fixation and interosseous wiring could lower the risk of infection from the exposed K-wire tip outside the skin with the intramedullary location of the entire length of the K-wire. The surgery is easy, does not require radiological control, and does not damage the arterial plexus. Additionally, it is possible to extract the internal fixator after bone union at an outpatient clinic. However, there is limited research comparing surgical technique in nonunion of DIP arthrodesis, so it is still controversial which surgical technique is better. To this end, our study devised an inlay technique with a cortico-cancellous olecranon bone graft by analyzing the advantages and disadvantages of several surgical methods. This technique resolves bone defects using olecranon bone grafts and improves stability by K-wire fixation on the bone stock. Even if the nonunion was confirmed by serial radiography, the internal fixator could be removed. Our study found that DIP arthrodesis using K-wire fixation with cortico-cancellous olecranon bone graft showed a 100% union rate and satisfactory clinical outcomes. There were no complications or infections after the removal of the internal fixations.

Bone stock, rather than the fixation method, was the greatest determinant of successful arthrodesis in these patients. Numerous sites have been used to harvest bone for grafting, and the most common bone graft donor site is the anterior iliac crest [18,19]. Owusu and Isaacs [6] used the iliac bone graft for the inlay technique in one digit with failed DIP arthrodesis. However, it is an invasive method for small bone defects (such as 15 mm in diameter or less), although the iliac bone graft has the advantage of harvesting sufficient cancellous bone [20]. Furthermore, general anesthesia and a second surgical site are typically required for iliac crest harvest, and ambulation can be problematic [18]. On the other hand, the olecranon graft has the important advantage of being performed under an axillary block and in the same surgical field [18,19]. The donor site is distant from important neurovascular structures so that dissection can be performed quite easily. The site can also be harvested with a small incision of less than 2 cm, which provides the advantage of minimal scarring [20,21]. Considering these advantages, we preferred bone grafting with olecranon and failed arthrodesis using an olecranon graft in all our cases. We believe that olecranon bone grafting is a better option for this operation that deals with small bone defect reconstruction in hand or upper extremity surgeries.

This study has a few limitations. First, this was a retrospective study. Therefore, it was difficult to perform direct comparisons with other studies. We used a uniform surgical technique and postoperative management to minimize enrollment bias. Second, we had few cases of failed arthrodesis because patients with DIP joint arthritis are rare compared with other hand diseases. Finally, we did not directly compare patients who underwent other surgical techniques. Insufficient evidence supports the inlay technique with a cortico-cancellous olecranon bone graft is better than other techniques. Despite several limitations, our study is the first to report clinical and radiologic outcomes using the inlay technique with cortico-cancellous olecranon bone grafts. It will aid in the selection of surgical techniques for failed arthrodesis.

## 5. Conclusions

The inlay technique with a cortico-cancellous olecranon bone graft for failed DIP joint arthrodesis showed excellent results in terms of bone union rates and functional scores. Therefore, patients who have confirmed nonunion of the DIP joint and persistent complaints of symptoms may be an adequate surgical indication with the bone graft inlay technique and internal fixation.

## Figures and Tables

**Figure 1 medicina-58-01442-f001:**
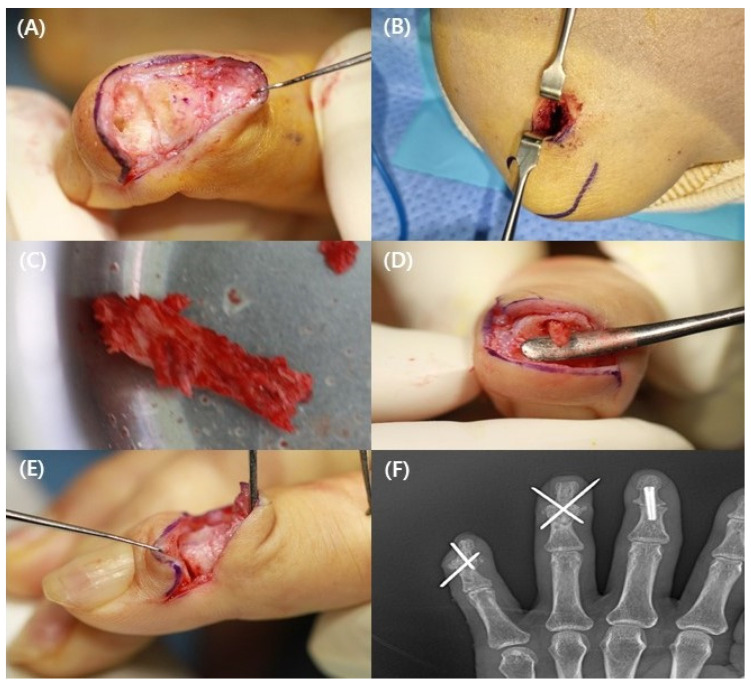
A 62-year-old woman underwent 3rd, 4th, and 5th distal interphalangeal joint (DIP) arthrodesis. (**A**) Two holes created by surgical burr within distal phalanx (DP) and middle phalanx (MP). (**B**) Bone grafting incision made at distal tip of olecranon. (**C**) The cortico-cancellous bone graft harvested from olecranon. (**D**) The bone peg impacted into hole created in MP with DIP joint flexion. (**E**) The distal aspect of bone peg impacted into DIP hole with DIP joint extension. (**F**) Postoperative plain radiograph fixated K-wires and inlay technique with cortico-cancellous bone graft.

**Figure 2 medicina-58-01442-f002:**
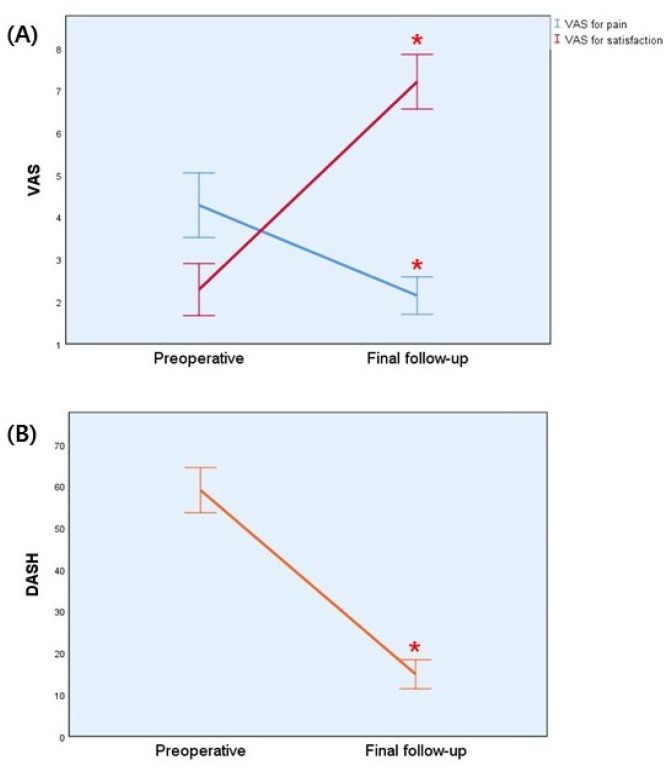
Graph showing functional outcomes at preoperative and final follow-up period; (**A**) VAS scores for pain (blue) and satisfaction (red) (**B**) DASH score (yellow). * *p* < 0.05. Error bars represent 95% confidence intervals. VAS: visual analog scale; DASH: Disabilities of Arm, Shoulder and Hand.

**Figure 3 medicina-58-01442-f003:**
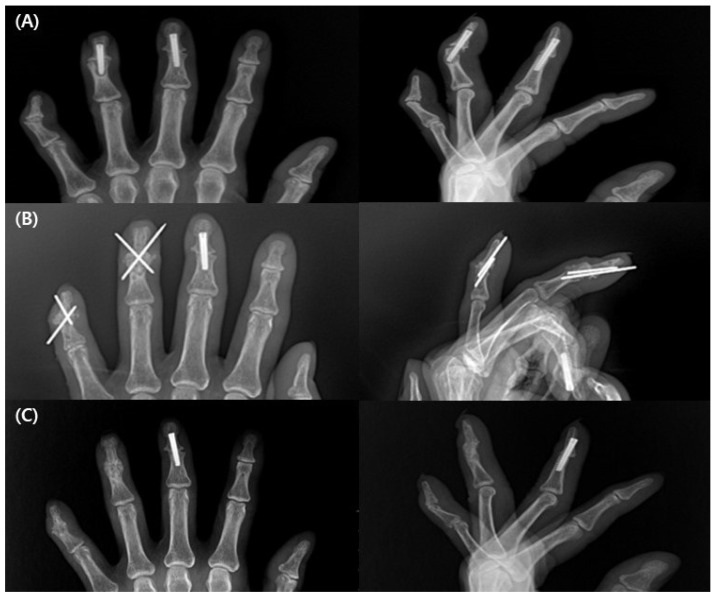
A 62-year-old woman underwent 3rd, 4th, and 5th distal interphalangeal joint (DIP) arthrodesis. (**A**) At 16 weeks after the initial surgery, pain and tenderness were sustained and radiograph shows no evidence of union with 4th and 5th DIP joint. The 5th finger was secured with K-wires, and the 3rd and 4th fingers were secured with Acutrak screws at initial surgery. (**B**) Post revision surgery with K-wires and inlay technique with cortico-cancellous olecranon bone graft. (**C**) The bony union achieved radiographically at 6 weeks after the revision surgery.

**Table 1 medicina-58-01442-t001:** Patient demographics.

No	Age	Sex	Affected Digit	First Treatment	Bone Graft Donor Site	Revision K-Wires Fixation	Interval (Weeks) to Reoperation	Primary Surgical Cause	Follow-Up Period (Weeks)
1	42	F	2nd	K	O	2	16	N	24
2			3rd	K	O	2			
3			5th	K	O	1			
4	57	F	5th	S	O	2	23	N	29
5	61	M	3rd	K	O	2	18	T	26
6	54	F	2nd	K	O	1	13	N	20
7			3rd	K	O	1			
8	65	F	1st	K	O	2	16	N	22
9	25	F	1st	S	O	1	20	I	28
10	69	F	4th	K	O	2	26	I	34
11	65	F	3rd	K	O	2	17	N	30
12	68	F	5th	K	O	1	20	N	30
13	59	M	1st	K	O	2	32	N	40
14	62	F	1st	S	O	2	16	T	33
15	55	F	5th	K	O	1	22	N	30
16	48	F	3rd	K	O	2	20	I	30
17			4th	K	O	1			
18	62	F	4th	S	O	2	16	N	32
19			5th	K	O	2			

M: male, F: female, 1st, 2nd, 3rd, 4th, and 5th refer to respective fingers, K: Kirschner wire fixation, S: Headless screw, O: Olecranon, I: Iliac bone, T: Traumatic, I: Inflammatory, N: Non-inflammatory.

**Table 2 medicina-58-01442-t002:** Clinical and functional outcomes.

	Preoperative	Final Follow-Up	*p*-Value
VAS for pain	4.29 (±1.33)	2.14 (±0.77)	0.001
VAS for satisfaction	2.29 (±1.07)	7.21 (±1.12)	0.001
Quick DASH score	59.09 (±9.38)	14.92 (±6.03)	0.001

Data presented as mean ± standard deviation. VAS: visual analog scale; DASH: Disabilities of Arm, Shoulder and Hand.

## Data Availability

Not applicable.

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
