# Peer review of "The Inlay Technique with Cortico-Cancellous Olecranon Bone Graft Used for Revision of Failed Distal Interphalangeal Joint Arthrodesis"

_medicina, 2022, doi:10.3390/medicina58101442_

Round 1

Reviewer 1 Report

This is the retrospective evaluation of the inlay bone grafting technique for failed arthrodesis of the DIP joint. Although DIP arthrodesis is a reliable surgical treatment, the nonunion rate is relatively high and additional surgery is required.  The authors evaluate the inlay bone grafting technique by selecting 20 cases from 2010 to 2020, describing its effectiveness. Although the study is well presented with the surgical procedure n Fig. 1 and the patient demographics in Table 1, the manuscript can be improved by responding to the following concerns.

Major

Rationale: it is stated that … no single technique has gained universal popularity. Therefore, we hypothesized that the inlay technique … This statement in the discussion section does not provide a rationale for selecting the inlay technique.

·         Other options: other options are suggested: various surgical techniques for the treatment of such nonunions have been … It is recommended to expand the comparison with other methods.

·        Small bone grafting with olecranon: Instead of using a qualitative phrase such as “small bone grafting with olecranon,” it is recommended to describe it quantitatively.

·        Table 2: the data in Table 2 should be presented in a bar chart.

Minor

·         Line 35: “15% to 20% percent” should read “15% to 20%”.

·         Line 129: “Interval.” This word should be removed.

Author Response

Thanks for good review. Critiques and comments from reviewers were greatly helpful. In response to reviewer 1, we have made changes based on your valuable comment. We hope these revised manuscript will convince all reviewers.

Our responses are written in red and the revised paragraphs in response to the comments are written in highlighted with yellow background under each comment.

Reviewer 2 Report

The paper is very interesting and presents the clinical outcomes following the use of inlay technique and would be of interest to the broad audience of the Medicina journal. 

The paper is well written, the results are clearly presented and properly discussed. However, I would suggest to expand the introduction section and add more information on the cortico-cancellous inlay technique in that section to improve the readability of the paper.  

Minor mistake: There are two ":" following Background and Objectives in the abstract

Author Response

Thanks for good review. Critiques and comments from reviewers were greatly helpful. In response to reviewer 2, we have made changes based on your valuable comment. We hope these revised manuscript will convince all reviewers.

Our responses are written in red and the revised paragraphs in response to the comments are written in highlighted with yellow background under each comment.
